# "If we lose it, we are worried": Individual and provider level perceptions towards weight change among people living with HIV who undergo TB screening in routine health care settings in Gauteng Province, South Africa

Tanyaradzwa Nicolette Dube[1]*, Fezeka Mboniswa[1], Samkelisiwe Ethel Qwana[2], Salome Charalambous[1,3], Nontobeko Ndlovu[1], Alison D. Grant[4,5], Yasmeen Hanifa[4], Johannes Machinya[6]

1 The Aurum Institute, Johannesburg, South Africa, 2 Research Consulting Specialists CC, Johannesburg, South Africa, 3 School of Public Health, Faculty of Health Sciences, University of the Witwatersrand, Johannesburg, South Africa, 4 TB Centre, London School of Hygiene & Tropical Medicine, London, United Kingdom, 5 Africa Health Research Institute, School of Laboratory Medicine and Medical Sciences, College of Health Sciences, University of KwaZulu-Natal, Durban, South Africa, 6 Department of Sociology, University of the Witwatersrand, Johannesburg, South Africa

* tndube@auruminstitute.org

## Abstract

### Background

HIV weakens the immune system, increasing the risk of tuberculosis (TB) in people with living HIV (PLHIV). People living with HIV and on antiretroviral treatment (ART) often experience physical body size changes. Studies have found a significant discrepancy between PLHIV's self-reported weight loss and their measured weight loss when being screened for TB using the WHO tool. To understand this inconsistency, a qualitative sub-study was conducted to explore perceptions and attitudes towards weight change among adults attending HIV care, as well as health care workers in public clinics in Gauteng, South Africa.

### Methods

Our qualitative study was nested within the XPHACTOR study. A total of seven focus group discussions were conducted, five with adult participants attending for HIV care and two with health care workers and research staff in clinics around Gauteng. Inductive thematic analysis was used to analyse the data.

### Findings

The majority of PLHIV preferred to gain weight due to fear of stigma associated with weight loss. Weight loss is associated with HIV/AIDS, suggesting that people attending HIV care may underreport weight loss in the context of a TB symptoms screening

**Data availability statement:** All relevant data are within the manuscript and its Supporting information files. The data analyzed for this manuscript has been provided.

**Funding:** ADG received the award Gates Foundation, grant number OPP1034523 https://www.gatesfoundation.org/ The funders did not play a role in the study design, data collection and analysis, decision to publish, or preparation of the manuscript.

**Competing interests:** The authors have declared that no competing interests exist.

tool because they fear stigma. Participants reported that weight changes impacted their daily lives and had psychological effects on them. Some PLHIV described lipodystrophy as disproportional weight gain. Culture and media have an influence on the perception of ideal body size and shape for both men and women.

## Conclusions

Underreporting weight loss might result in poor sensitivity of the WHO TB screening tool and suggests that we need either alternative ways to determine weight loss or screening tools for TB that are less dependent on reported symptoms.

## Background

Tuberculosis (TB) is one of the most common opportunistic infections in people living with the human immunodeficiency virus (PLHIV) and a major contributor to morbidity and mortality [1,2]. The risk of developing active TB disease is estimated to be 16 (uncertainty interval 14–18) times greater in PLHIV than those without HIV infection [3]. Weight loss is a common symptom of TB, and the World Health Organization (WHO) recommends screening people living with HIV for TB using a tool that comprises questions on cough, fever, night sweats, and unintentional weight loss [4]. Unintentional weight loss is a cardinal symptom of TB and reporting of weight loss is a key entry point to testing for TB, so understanding perceptions and reporting of weight loss is important with regards to finding people with active TB.

People living with HIV and on antiretroviral treatment (ART) often experience physical body size changes such as weight change, and other body shape changes that can lead to stigmatization [5–10]. In South Africa, changes in body shape were particularly common among PLHIV taking stavudine, which was routinely used in ART between 2004 and 2010 but is now rarely used because of its side effect profile [11].

Two studies conducted in South Africa found a mismatch between PLHIV's self-reported and measured weight change: one was a quantitative study and the other conducted in-depth interviews focusing on adolescents only [12,13]. It is important to understand perceptions of PLHIV regarding weight and body shape changes because these may impact responses to the weight loss component of the WHO TB symptom screening tool, as well as influence adherence to ART and TB treatment. A better understanding of weight perceptions may help improve TB screening and guide policies and recommendations for supporting PLHIV when they experience weight and body shape changes.

We conducted an in-depth qualitative study to understand weight perceptions among adults living with HIV as part of the XPHACTOR study. XPHACTOR was a cohort study investigating TB diagnostic pathways among adults attending routine HIV care in South Africa between 2012 and 2014 when standard ART regimens in South Africa included stavudine [14–16]. In XPHACTOR, PLHIV were screened for TB using the WHO screening tool and significant discrepancies were noted between participants' self-reported weight loss and their measured weight loss. To understand

this inconsistency, a qualitative sub-study was conducted to explore perceptions and attitudes towards weight change among adults attending HIV care, as well as among health care workers (HCWs) in public clinics in Gauteng, South Africa. We sought to investigate whether attitudes to weight and body size might affect how people responded to the questions in the WHO TB symptom screening tool and whether people with stavudine-related lipodystrophy might describe the change in body shape as weight loss.

## Methodology

### Study design

This was an exploratory qualitative sub-study.

### Study setting and population

Focus group discussions (FGDs) were conducted at public clinics providing care for PLHIV in Gauteng. Participants were adults attending the clinics for routine HIV care at XPHACTOR study sites, HCWs, and XPHACTOR research staff.

### Data collection

Seven FGDs with a semi-structured guide were conducted: five FGDs with purposively sampled adults attending for HIV care and two FGDs with HCWs and research staff. The FGDs with PLHIV included two groups with male-only participants, two groups with female-only participants, and one mixed group. For HCWs, there were two FGDs, one FGD was done with doctors and dieticians and the other with other HCWs and research staff. All the FGDs were conducted at the clinics where the study was being conducted and where PLHIV and health workers were attending or working. Adult PLHIV who were not XPHACTOR study participants were approached in-person for participation in the study via an ART support group linked to the clinic. Research Assistants and HCWs were contacted via email. The FGDs were conducted face-to-face by an experienced independent female research consultant (SEQ) with a Masters' degree and was assisted by a co-facilitator with a bachelor's degree. All participants completed the FGD. To ensure that important points were discussed, the moderators led the discussions using FGD guides for PLHIV (S1 File) and HCWs and research staff (S2 File). There was no pre-existing relationship between the moderators and participants. FGDs were conducted within the clinic setting where participants collect their medication.

The length of the FGDs ranged between 83 minutes and 202 minutes. All sessions were audio recorded. The moderators took some field notes. No one else was present in the FGDs besides the participants and researchers. We initially planned to conduct around three FGDs with about eight to twelve participants in each group but eventually conducted seven FGDs to reach saturation. The number of discussions increased because we separated male and female FGD's (two groups per gender) plus a mixed gender discussion to capture sufficient diversity in perspectives and experiences. Sample size per group was determined by availability of participants and previous qualitative research which recommends six to twelve participants per FGD [17,18]. This range ensures sufficient diversity, a balanced interaction to allow all participants to contribute as well as effective facilitation of the discussion by the moderators. There were no repeat interviews done. All participants in the FGDs except those who were members of the research team were given ZAR 100 = USD 15 to compensate for their time.

In addition to using a moderator guide, the interviewer probed on the questions and was able to gather more detailed responses from the participants. Participants were asked questions about their perceptions on ideal body weight, what informed these perceptions, personal experiences of, and attitudes towards body weight changes. In addition, they were asked questions about causes of weight change and reasons why weight changes measured at the clinic may differ from those reported by the patient. The HCWs and research staff were also asked questions about recent experiences of consultations with PLHIV around body weight. Images of body figures of different body shapes developed by Stunkard

were used to explore perceptions about body size and body image. The figures ranged from one to nine where one is the thinnest body type and nine is the largest type [19]. There were separate figures for males (Fig 1) and females (Fig 2). Participants were shown the images and were asked which body image represented them, which body image they preferred, and the most attractive body image.

Discussions were conducted in local languages IsiZulu, Sesotho, Setswana, and English. All discussions were tape-recorded to ensure that all the points were captured. All interviews were transcribed and interviews done in local languages were translated to English. Translations and transcriptions were done by experienced researchers fluent in the study languages. Transcriptions were done verbatim to capture nuances and deliberations including non-verbal cues such as short silences and laughs. Quality assurance was conducted whereby a second researcher checked the accuracy of the transcripts against audio recordings.

### Ethical considerations

The study protocol was approved by the University of the Witwatersrand Human Research Ethics Committee, Johannesburg (M 120343); the University of Cape Town, Cape Town Human Research Ethics Committee, South Africa (106/2012); and the London School of Hygiene & Tropical Medicine Research Ethics Committee UK (6165). All participants were informed about the study and gave written consent to participate and be recorded during the FGDs. Audio recordings, field notes, and transcripts were identified only by a unique study number. All study records were stored securely in locked filing cabinets where access to the records was restricted to specified study team members.

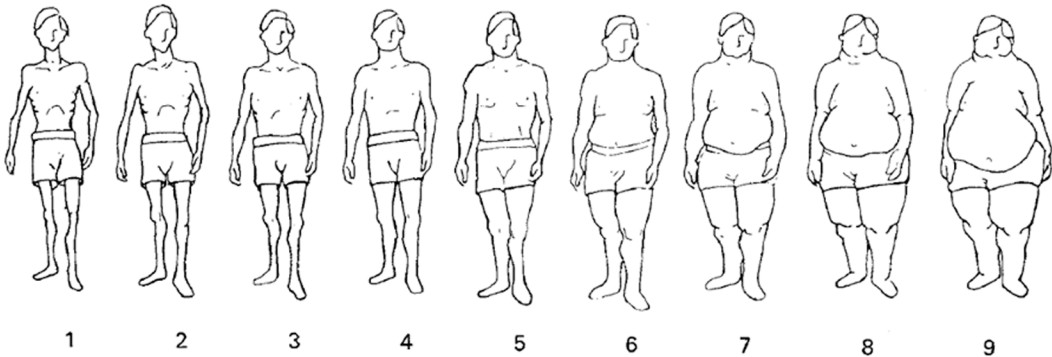

**Fig 1. The Stunkard scale of male body images shown to participants [19].**

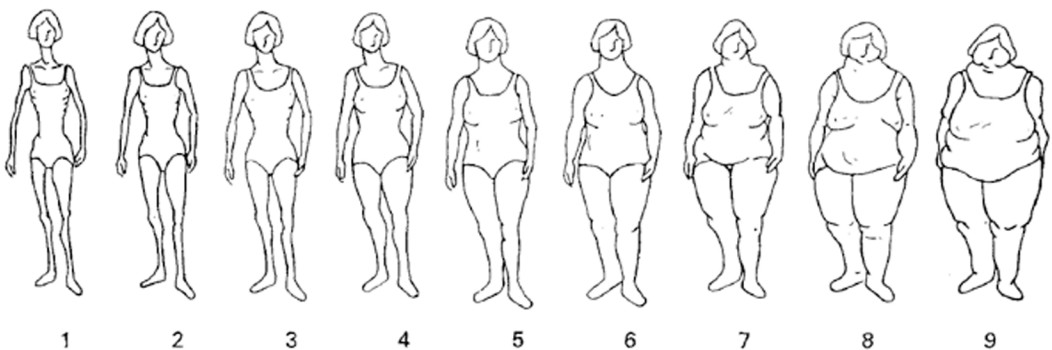

**Fig 2. The Stunkard scale of female body images shown to participants [19].**

## Data analysis

A codebook was developed by two researchers (TD and FM) using inductive coding. Data were analysed manually using Excel. The two researchers independently inductively coded the transcripts and agreed on the emerging themes. Two data coders manually coded the data (TD and FM). TD has a Masters' degree and FM has an Honours degree. The senior author is an experienced qualitative researcher with a doctoral degree and supervised the data analysis (JM). The data coders manually coded a sample of the transcripts and the senior author (JM) qualitatively and quantitatively checked intercoder reliability between the coders. Inductive thematic analysis was employed to develop themes. Both males and females were included to account for variation in perceptions by gender. We used an Excel spreadsheet to monitor the number of steps that were followed during the manual coding process. TD, FM and JM reviewed and refined emerging themes repeatedly during analysis until inductive thematic saturation was reached, where no new themes or categories emerged from the data [20]. Transcripts and findings were not returned to participants for comment. The COREQ (Consolidated criteria for REporting Qualitative research) checklist was used to check the content of this manuscript (S3 File).

## Results

Seven FGDs (labelled 1–7) were held between 6th and 21st August 2014, with 7–12 participants in each focus group. The five adult PLHIV FGDs (two males, two females and one mixed gender) had 7–12 participants in each group. The HCWs and research staff FGD comprised of ten participants (seven routine HCWs and three research staff). The doctors and dieticians FGD had seven participants (five doctors and two dieticians) who were all routine staff.

We identified three main themes and five sub themes. The main themes were: 1) perceptions and attitudes towards weight change; 2) ideal body size and shape; 3) negative impact of weight loss on participant's daily life. The sub themes were i) perceptions towards weight change; ii) attitudes towards weight loss; iii) stigma and loss of social support; iv) psychological impact of weight loss and v) changes in dressing due to weight changes.

### Main theme 1: Perceptions and attitudes towards weight change

**Sub-theme (i): Perceptions towards weight change.** Participants in all the FGDs for adult PLHIV attending for HIV care expressed high level of awareness of the possible physical body changes experienced by PLHIV and those on ART such as weight loss, weight gain and associated changes to body shape. The participants were aware of the main causes of weight loss, and they linked weight loss to poor diet, stress, active TB, HIV/TB treatment side effects, and non-adherence to HIV/TB treatment. It was reported that denial or failure to accept and embrace one's HIV positive status also resulted in weight loss.

*I am saying, another thing that can make a person to lose weight is that because I am HIV positive, I must check whether I have TB or not because that also causes a person to lose weight (FGD 3, Females).*

Participants in the separate groups were asked how they knew that they were losing weight. Both male and female participants reported that they noticed that they had lost weight if their clothes were not fitting well.

HCWs and research staff indicated that weight loss could be caused by other health conditions among PLHIV. As such, when they noted weight loss in a PLHIV attending for HIV care, they reported that they would usually or ideally check the PLHIV's viral load, CD4 count, investigate for TB and other opportunistic infections, screen for cervical cancer, and test for diabetes.

Both PLHIV and HCWs identified ART side effects as one of the causes of weight gain and change in body shape among PLHIV:

*Let me say, you can see that I am slender but if I take these tablets, you see and acquire a big stomach, they will then say he is in this condition, he is now taking these tablets, isn't these tablets are making him to have a big stomach (FGD 1, Males).*

Participants from the PLHIV's FGDs were happy to gain weight. They highlighted that weight gain was a sign that they were adhering to HIV treatment and that they were responding well to treatment:

*… if we gain weight, I believe that we are happy, because it means that no, we are happy, we are eating well, we are doing things the right way and that our medication is working well but if we lose it, we are worried because it's like we are not taking these tablets* (FGD 5, Mixed).

PLHIV also perceived weight gain as a sign that one has accepted and embraced their HIV positive status:

*It means that you are relaxed you are not obsessing over your HIV status, you see, you do not put yourself under stress, you think that I do have tablets. If I take tablets everything is okay* (FGD 1, Males).

Some PLHIV described lipodystrophy as disproportional weight gain. While PLHIV indicated that they would prefer weight gain to weight loss, they highlighted that sometimes when one gains weight, it might not be proportional to one's body size because they will gain weight on the upper body and lose weight on the lower part of the body, and this worried them:

*You feel good when you gain, but what you do not like is that sometimes you can gain on the upper part and lose weight on the lower part. So that is what makes us sad* (FGD 4 Females).

**Sub-theme 2: Attitudes towards weight loss.** There were both positive and negative attitudes towards weight loss. Some participants revealed that excessive weight gain was not healthy, so they adopted a healthy lifestyle including regular exercise and celebrated weight loss after gaining excessive weight, as shown in the quote below:

*I was happy today because I had lost a lot of weight* (FGD 3, Females).

The majority of the participants demonstrated negative attitudes toward weight loss and some associated weight loss with the side effects of ART. Such participants reported that they would consider stopping ART to gain weight:

*For me sometimes, I get a feeling that maybe these ARVs do not get along with my system and then have another feeling that hey I will stop taking them since I do not see its results, I keep losing weight, that is why most of the time you find that after people have started ARVs.., you find that a person does not take them anymore and they will say, if I don't take them I regain my weight,* (FGD 4, Females).

Participants further reported that there were some nurses who reproached them for delaying seeking health care when they visited the clinic with sickness and showing signs of weight loss:

*You see at the clinic, the nurses scold at you if you visited them when you were really too sick and badly affected, [they] can see that you have lost weight, the nurse scolds at you and say, you are this and that why did you not come and visit me (says P mimicking a female nurse) they scold at you there.* (FGD, 1 Males).

The HCWs reported that they would become inquisitive about adult PLHIVs lifestyles if they showed signs of gaining weight as highlighted by a dietician in the HCWs group:

*If they are gaining weight, we want to know if they are exercising or not* (FGD 7, Dieticians and doctors).

In addition to weighing PLHIV on a scale, HCWs stated that they used alternative ways to identify if PLHIV had lost weight without openly using the words "weight loss" to get accurate answers and avoid being seen as stigmatizing PLHIV. The HCWs reported that they would instead ask participants if the size of their dress has changed or if family members had commented on changes in their body shape:

*'Has your dress size changed?' could be the alternative question* (FGD 7, Dieticians and doctors).

*And then you can ask sometimes, if anyone, your friend, or family complained that your body shape has changed* (FGD 7, Dieticians and doctors).

### Main theme 2: Ideal body size and shape

The participants were aware that weight change also alters both body size and shape. Participants were shown both male and female Stunkard body image figures during FGDs to elicit their perceptions on ideal body size and shape [19]. The majority of male participants selected body image figure number five in the middle as the ideal figure for men (Fig 1). Some men suggested that women should be the same size as men and others reported that women should be slightly bigger because they bear children. Female participants selected body image figure number six as the ideal body type because the image had a mid-sized proportional body (Fig 2). The choice of ideal body size was influenced by media for both women and men as indicated in the responses below:

*I think influence in terms of being muscular has a lot to do with media. TV gives you the suggestion that someone who is muscular is more acceptable to the community* (FGD 2, Males).

*they make jokes out of us, as women, they say check her out she has unnecessary fats on the waist area, you know it would have been better if I found someone like Beyoncé [female celebrity], you see.* (FGD 4, Females).

Participants further reported that culture also influenced how people perceive ideal body shape especially for women.

*My sister is right about culture, Zulus prefer fat women. They say she is well-fed* (FGD 3, Females).

*And I think we have a black culture that a black woman has to be round (Participants agreeing with one another), they will just be worried, when number 4, they won't even look at you* (FGD 6, HCW and Research Staff).

HCWs revealed they would ask PLHIV who had signs of lipodystrophy whether other people had commented on the changes in their body shape or size. According to the HCWs, PLHIV were more worried about changes in body shape or size and not so much about weight loss. It was uncommon for PLHIV to visit the clinic because they were worried about weight loss, but they usually visited the clinic when they noticed changes in body shape:

*What I have noticed is that it is mostly PLHIV who have got lipodystrophy, they are the ones who are more worried about their weight that's one thing I have realised because they are saying, no, I no longer have my buttocks, they are gone, my legs are now thinner.* (FGD 7, Dieticians and doctors).

HCWs stated that when they noticed changes in body shape and size among the PLHIV, they changed the regimen for those who were taking stavudine. PLHIV on the second-line ART regimen (at the time of the study, 2014, azidothymidine, lamivudine, lopinavir/ritonavir), were counselled and some were referred to government-funded plastic surgery services. HCWs also encouraged PLHIV to exercise and improve their diet.

FGDs with HCWs and research staff revealed that PLHIV who lost weight were referred to dieticians, social workers, psychologists, and religious leaders for support. PLHIV with socio-economic challenges were referred to the South African Social Security Agency for the Disability Grant. HCWs also revealed that they involved the families of PLHIV experiencing weight loss to explore potential causes.

**Main theme 3: Negative impact of weight loss on participant's daily life**

**Sub-theme 1: Stigma and loss of social support.** Participants reported that changes in physical body size, particularly weight loss, resulted in them being stigmatized in the community. Some of the participants reported that they experienced body shaming, name-calling, gossip, and discrimination because they had lost too much weight. Participants in the FGDs lamented that people in the communities would quickly conclude that a person has AIDS once one begins to lose weight. The participants reported that those who show signs of weight loss are said to have bought OMO (washing powder) because the HIV "washed" way the person's body the way OMO washes dirt:

*They will say, so and so bought OMO can you actually see so and so, so in that manner they call you by those names when you are on the passing by, and you would think my goodness. They do this so that you cannot understand you see, you may not know why you are regarded as having bought OMO, when you look around no one bought OMO, they regard you as that since you lost weight* (FGD 1, Males).

Participants with lipodystrophy were stigmatized and were called names such as "high heels" and KFC because of having thin legs and a fat upper body:

*Thin legs. Do you know what a KFC drumstick looks like? When they see you, they say: It's KFC! She went to buy KFC* (FGD 3, Females).

Participants also reported stigmatizing attitudes from both community and family members which include being ignored and people keeping some distance from them:

*Mostly, if they know that you are positive through appearance, the loss of weight, they start ignoring you and mostly the names like killer* (FGD, 2 Males).

*Even your family still does that thing, they say hey, don't stand too close to me you will infect me with this thing* (FGD 4 Females).

Some participants in the FGDs reported that PLHIV who show signs of weight changes may lose social support from their families and they attributed this to a lack of adequate knowledge on HIV:

*So, this is what happened …, my mother is an elderly person so she does not know the difference between HIV and AIDS. When my brother took a cup and drank from it, she would want nothing to do with that cup. So, I called her and tried to teach her about what HIV and AIDS are. I said I'm asking you to please support him* (FGD 3, Females).

**Sub-theme 2: Psychological impact of weight loss.** The participants in focus group discussions revealed that when some PLHIV discover that they have lost weight, they become so stressed:

*The thing is, you just become stressed out, you just cannot concentrate on the things that you are doing at home, do you understand, you become short-tempered, you ask yourself what is going on, what did I do, what mistake did I make in order for me to lose weight in this manner* (FGD, 4 Female).

Some PLHIV revealed they feel anxious and stressed when they are asked to stand on the scale and be weighed at the clinic:

*Before standing on the scale, I was afraid that they are about to weigh me now and it's possible that I may have lost weight* (FGD1, Males).

**Sub-theme 3: Changes in dressing.** When they notice that they have lost weight, participants reported that they would change the way they dress to cover signs of weight loss:

*So, now that I have lost weight, when I get dressed and wear something that hides my bums they say why are you making yourself a granny? You must wear the latest fashion, and I say: hey I am not a child (FGD, 3 Female).*

## Discussion

Participants demonstrated a strong awareness that some physical body changes such as weight gain and weight loss may be symptoms of HIV and AIDS. The participants further indicated that negative weight changes, i.e., weight loss, attract stigmatizing attitudes from society and that stigma remains a huge concern for PLHIV. As a result, the participants preferred gaining to losing weight. These findings are consistent with findings from other studies in South Africa that found that PLHIV would rather gain weight than be thin and have people assume that they are infected with HIV [5,12,21]. The results obtained in our study were similar to the findings of a qualitative Ugandan study that found that women living with HIV did not want to lose weight as this could lead to people suspecting that they were HIV positive. Women living with HIV who lost weight in the same study also reported that they felt stigmatized, and it affected their adherence to ART [21]. The finding that it was rare for PLHIV to seek healthcare services because of weight loss concerns may negatively affect efforts to screen for and diagnose TB early. Several studies have found an association between weight loss and delay in TB diagnosis [22–24]. A quantitative study consisting of 185 people with TB in KwaZulu Natal South Africa found that weight loss was associated with delayed care [22]. The researchers suggested that the stigma of weight loss because of its association with other diseases could have been a key factor contributing to this delay [22].

In our study, HCWs reported that because of the stigma attached to weight loss, they avoid asking PLHIV direct questions about weight loss because the fear of the stigma associated with weight loss may hinder PLHIV from telling the truth. The stigma associated with weight loss may make the WHO screening tool less sensitive, specifically the question on weight loss. A potential alternative to asking questions about weight loss when PLHIV present for treatment of check-ups is to measure their weight at the visit and compare this to previous weight measures recorded in their file. The stigma associated with HIV and TB needs to be addressed for example by educating PLHIV, family members, and the community on HIV and TB. The education can be done through community outreach campaigns using tool kits that have been developed to address TB-related stigma [25–27].

Despite the preference to gain rather than lose weight amongst the majority of participants, some participants were aware that excessive weight gain has repercussions and reported adopting a healthy lifestyle including regular exercise and celebrated weight loss.

A mixed method study including 513 adult women was conducted in Cape Town, South Africa to explore the perception among black South African women that people who are thin are infected with HIV or have AIDS. The study found that participants preferred to be overweight and at risk of acquiring cardiovascular diseases, rather than to lose weight and be stigmatised as infected with HIV [5]. It is important to educate PLHIV that excessive weight gain may lead to obesity and increase the risk of non-communicable diseases. Therefore, PLHIV should be encouraged to exercise and eat well to reduce the risk of these diseases [8,13,28].

The psychological impact of weight changes leading to non-adherence found in our study has been reported in other studies [8,13,21,28]. Lipodystrophy is a concern among PLHIV. The finding that changes in body image affect PLHIV psychologically, with some considering stopping treatment, is consistent with other studies. A cross-sectional quantitative study on lipodystrophy among PLHIV in Tanzania found that changes in body shape may affect PLHIV psychosocial function, quality of life and may lead to PLHIV considering stopping treatment [28].

Our study found that fuller women were considered to have the ideal body shape. This finding is consistent with findings from other previous studies [5,13]. A qualitative study conducted among 18 adolescents in Durban using in-depth interviews revealed that female respondents desired to be thick and curvaceous, and males desired to be bigger and muscular, and this was linked to the influence of cultural norms on understanding of an ideal body shape [13]. Similarly, a study conducted in Cape Town found that cultural beliefs appeared to have a significant influence on the women interviewed, despite more than half of the participants having a high school education. Some women in this study reported that women are culturally expected to have a fuller figure [5].

Stavudine is now rarely used and so few PLHIV should newly develop lipodystrophy. Those with established lipodystrophy should have been switched to a different regimen which does not cause lipodystrophy, but these people may need support and reassurance to continue their current ART [11].

Participants in our study reported that changes in dress sizes made them realize they had lost weight. HCWs in our study recommended that asking participants about changes in clothing size was more useful than directly asking about unintentional weight loss. Asking clothing size is a self-report measure which can get biased responses, but we feel it is more acceptable to patients rather than directly asking about weight loss. A case cohort quantitative study in the Netherlands found that self-reported clothing size appeared to predict cancer risk independently of body mass index, suggesting that self-reported clothing size can be a useful measure to consider in epidemiologic studies [29].

Our study findings are significant as TB remains a leading cause of death among PLHIV so understanding their perceptions around weight change and how this may affect TB symptom screening is important. In addition, since this work, the new first-line treatment regimen for PLHIV includes dolutegravir, which has been found to have weight gain as a side effect. Hence, perceptions on weight gain highlighted in our paper remain relevant [21,30].

## Strengths and limitations

One of the strengths of this study is that we conducted FGDs with both males and females and managed to get perceptions on weight change from both genders. However, the study was not without limitations. The sample size of the PLHIV who were interviewed was small and may not be representative of all PLHIV at the facilities. Again, PLHIV may not have been able to disclose sensitive personal information in focus groups. Despite these limitations, we managed to gain insights from PLHIV as well as HCWs on people's perceptions of weight change among people who live with HIV. The potential bias was minimised by the fact that the facilitator was an independent researcher, therefore would have been perceived to have adopted a neutral stance on the issues.

## Conclusion

The majority of PLHIV preferred to gain weight due to fear of stigma associated with weight loss. Weight loss is associated with HIV/AIDS, suggesting that people attending for HIV care may underreport weight loss in the context of a TB symptom screening tool. This would result in reduced sensitivity of the WHO TB screening tool. Our findings suggest that we need either alternative ways to determine weight loss, or screening tools for TB which are less dependent on reported symptoms. Furthermore, the policy implications of our paper suggest the importance of strengthening innovative TB testing strategies such as Targeted Universal Test and Treat (TUTT) which tests high risk groups including PLHIV irrespective of reporting TB symptoms or ART status [31].

## Supporting information

**S1 File. FGD guide for PLHIV.**
(DOCX)

**S2 File. FGD guide for HCWs and research staff.**
(DOCX)

**S3 File. COREQ checklist.**
(PDF)

**S4 File. Transcripts.**
(ZIP)

## Acknowledgments

The authors are grateful to the Department of Health nurses and staff at the facilities, the XPHACTOR research staff, and all the participants.

## Author contributions

**Conceptualization:** Salome Charalambous, Alison D. Grant, Yasmeen Hanifa.

**Formal analysis:** Tanyaradzwa Nicolette Dube, Fezeka Mboniswa, Johannes Machinya.

**Funding acquisition:** Alison D. Grant.

**Investigation:** Salome Charalambous, Alison D. Grant.

**Methodology:** Samkelisiwe Ethel Qwana, Salome Charalambous, Alison D. Grant, Yasmeen Hanifa.

**Project administration:** Nontobeko Ndlovu, Yasmeen Hanifa.

**Resources:** Salome Charalambous, Alison D. Grant.

**Supervision:** Salome Charalambous, Alison D. Grant.

**Validation:** Salome Charalambous, Alison D. Grant, Johannes Machinya.

**Writing – original draft:** Tanyaradzwa Nicolette Dube.

**Writing – review & editing:** Tanyaradzwa Nicolette Dube, Fezeka Mboniswa, Samkelisiwe Ethel Qwana, Salome Charalambous, Nontobeko Ndlovu, Alison D. Grant, Yasmeen Hanifa, Johannes Machinya.

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
