## [Decision Letter · Decision Letter 0]

27 May 2024

PONE-D-24-15618Perceptions and attitudes towards weight change among adults attending HIV care and healthcare workers in public clinics in South Africa.PLOS ONE

Dear Dr. Dube,

Thank you for submitting your manuscript to PLOS ONE. After careful consideration, we feel that it has merit but does not fully meet PLOS ONE’s publication criteria as it currently stands. Therefore, we invite you to submit a revised version of the manuscript that addresses the points raised during the review process.

**The work submitted in noted. The authors should add the significance of the findings and practical implication to current times. the data is old so please justify, especially noting the chronicity of HIV in South Africa. Reviewer 2 raised important comments to be addressed. **

We look forward to receiving your revised manuscript.

Kind regards,

Prof Y Malele-Kolisa, BDS, MPH, MDent, PhD

Academic Editor

PLOS ONE

Journal Requirements:

2. In the online submission form you indicate that your data is not available for proprietary reasons and have provided a contact point for accessing this data. Please note that your current contact point is a co-author on this manuscript. According to our Data Policy, the contact point must not be an author on the manuscript and must be an institutional contact, ideally not an individual. Please revise your data statement to a non-author institutional point of contact, such as a data access or ethics committee, and send this to us via return email. Please also include contact information for the third party organization, and please include the full citation of where the data can be found.

3. Please upload a copy of Figure 4 and 5, to which you refer in your text on page 12 and 18. If the figure is no longer to be included as part of the submission please remove all reference to it within the text.

Additional Editor Comments:

The work submitted is noted. Could the authors address the constructive Reviewer comments? Discuss the significance of the study findings in today's terms and add the relevant practical and policy implications

Reviewers' comments:

Reviewer's Responses to Questions

**Comments to the Author**

1. Is the manuscript technically sound, and do the data support the conclusions?

Reviewer #1: Yes

Reviewer #2: Partly

2. Has the statistical analysis been performed appropriately and rigorously? 

Reviewer #1: N/A

Reviewer #2: I Don't Know

3. Have the authors made all data underlying the findings in their manuscript fully available?

Reviewer #1: No

Reviewer #2: No

4. Is the manuscript presented in an intelligible fashion and written in standard English?

Reviewer #1: Yes

Reviewer #2: Yes

5. Review Comments to the Author

Reviewer #1: The manuscript is well written and the findings well supported, particularly with the quotes used, portraying clear and well selected thematic areas.

However, the aspect of weight loss in relation to the tool used for screening tuberculosis seems to appears only at the background. Considering the relationship between HIV/AIDS and TB, as well as the high chances of people living with HIV/AIDS to contract tuberculosis due to the compromise of their immune system, it would be nice to have it appear up front even from the title level. This will certainly give a sense of concern to the people living with HIV/AIDS and the "why we should worry" about their perceptions and attitudes on any weight change particularly the weight loss.

Lastly, on data availability and restrictions, if the author indeed de-identified the transcripts with a unique study number, then there should be no fears of bridging confidentiality if made public.

Thank you.

Reviewer #2: Dear Researchers

Well done on his work. I have made a number of comments directly onto the text of the draft and this is attached. please address these

6. PLOS authors have the option to publish the peer review history of their article (what does this mean? ). If published, this will include your full peer review and any attached files.

**Do you want your identity to be public for this peer review?** For information about this choice, including consent withdrawal, please see our Privacy Policy .

Reviewer #1: **Yes: ** Hellen Jepngetich Keny

Reviewer #2: **Yes: V** eerasamy Yengopal

---

## [Author Response · Author response to Decision Letter 1]

25 Nov 2024

Editors Comments

Response: Agreed. This has been done.

2. In the online submission form you indicate that your data is not available for proprietary reasons and have provided a contact point for accessing this data. Please note that your current contact point is a co-author on this manuscript. According to our Data Policy, the contact point must not be an author on the manuscript and must be an institutional contact, ideally not an individual. Please revise your data statement to a non-author institutional point of contact, such as a data access or ethics committee, and send this to us via return email. Please also include contact information for the third party organization, and please include the full citation of where the data can be found.

Response: Agreed. Data has been submitted with the revised manuscript.

3. Please upload a copy of Figure 4 and 5, to which you refer in your text on page 12 and 18. If the figure is no longer to be included as part of the submission please remove all reference to it within the text.

Figures with Stunkard body image figures have been added to the manuscript (Fig1 and Fig 2). The figures have male and female body image figures ranging from 1 to 9 and participants were choosing the image they preferred.

Response: Done

5. The work submitted is noted. Could the authors address the constructive Reviewer comments?

Discuss the significance of the study findings in today's terms and add the relevant practical and policy implications the data is old so please justify, especially noting the chronicity of HIV in South Africa

Response: Our study findings are significant in today’s terms as South Africa still has a high estimated HIV prevalence rate of 17.8 among adults (15-49 years) and 75% of PLHIV are on ART(23). TB remains a leading cause of death among PLHIV so understanding their perceptions around weight change and how this may affect TB symptom screening is important. In addition, since this work, the new first-line treatment regimen for PLHIV includes dolutegravir, which has been found to have weight gain as a side effect. Hence, perceptions on weight gain highlighted in our paper remain relevant (18,24).

Page 20 line 431-436

The policy implications of our paper suggest that in addition to TB screening, we should continue using innovative TB testing strategies such as Targeted Universal Test and Treat (TUTT) which tests high risk groups including PLHIV irrespective of reporting TB symptoms so that PLHIV with TB are not missed due to inaccurate self-reporting on TB symptoms. See line 458 page 20

Reviewer 1

The aspect of weight loss in relation to the tool used for screening tuberculosis seems to appear only at the background. Considering the relationship between HIV/AIDS and TB, as well as the high chances of people living with HIV/AIDS to contract tuberculosis due to the compromise of their immune system, it would be nice to have it appear up front even from the title level. This will certainly give a sense of concern to the people living with HIV/AIDS and the "why we should worry" about their perceptions and attitudes on any weight change particularly the weight loss.

Response: Agreed. The title has been revised to include TB screening among PLHIV.

“If we lose it, we are worried”: Individual and provider level perceptions and attitudes towards weight change among people living with HIV screening for tuberculosis in routine health care settings in Gauteng Province, South Africa. Also the risk of developing TB among people living with HIV was included on page 4 (line72-73) in the background

2. Lastly, on data availability and restrictions, if the author indeed de-identified the transcripts with a unique study number, then there should be no fears of bridging confidentiality if made public.

Response: Agreed. Data has been submitted with the revised manuscripts

Reviewer 2

1. this is misleading. your population comes from one part of the country (Gauteng) ..so your tiltle should reflect this. Are these findings generalizable ..then i wont have a problem

Response: Agreed. The title has been changed to reflect Gauteng Province

“If we lose it, we are worried”: Individual and provider level perceptions and attitudes towards weight change among people living with HIV screening for tuberculosis in routine health care settings in Gauteng Province, South Africa.

2. Line 125

Research Assistants and HCWs were conducted via email

do you mean contacted? If not, then this sentence does not make sense!!

Response: Yes. We meant contacted. This has been edited on page 6 line 126

3. Quote on line 236:

When I thought about the scale I was so excited because I have been weighing 152, so I had to hit the gym whatever it took. So today when I thought about coming to the scale I was excited. I wanted to see if this 152 had gone down. ((They all laugh)). When I got there I found out that it had gone down it’s 62.5. and I was happy (FGD 3, Females).

Reviewer’s comment: not sure if this is kgs ..also the drop in weight then is significant!

Response: Yes, the participant meant kg’s. We agree that the weight loss reported her was massive and we quoted a participant who may have gotten the figures wrong. We have decided to use a different quote to support the finding that other participants had a positive attitude towards weight loss on page 12 line 244

I was happy today because I had lost a lot of weight (FGD 3, Females).

4. This sentence contradicts what you have been saying all along that PLHIV under-report weight loss but here you are saying that they report weight loss even though they have not lost weight!!!

Response: We agree with the reviewer that the data primarily support the idea that PLHIV are likely to underreport weight loss and have removed the two sentences in the discussion that talk about overreporting.

5. So maybe somewhere in your manuscript you need to highlight the probability of TB diagnosis in PLHIV.

also is weight loss a function of the drug or a symptom of TB in these patients or both?

Response: The risk of developing active TB disease is estimated to be 16 (uncertainty interval 14–18) times greater in PLHIV than those without HIV infection. (WHO, 2023). The statement has been added to the background line 72-73.

Line 198-204 highlight participant’s perceptions on possible causes of weight loss which includes side effects of medication and TB disease. Line 246 refers to the idea that ART might cause weight loss. A quote has been added to support the statement regarding TB causing weight loss.

“I am saying, another thing that can make a person to lose weight is that because I am HIV positive, I must check whether I have TB or not because that also causes a person to lose weight” (FGD 3, Females).

6. Line 386 Statement: A potential response could be to weigh PLHIV instead of asking them the question on weight loss

Comment: Surely this applies to anyone from this community regardless of their HIV status?

Response: Agreed. This has been changed to: A potential response could be to weigh people seeking health care services instead of asking them the question on weight loss. (Lines 388-389) page 18

7. Statement: It is important to educate PLHIV that excessive weight gain may lead to obesity and increase the risk of non-communicable diseases

Comment: surely here a better measure would be BMI. the researchers had the opportunity to weigh the participants. i am not sure if the height was taken. if so a BMI could have been calculated and then the quest of excessive weight gain would have been easy to recognize in the context of a BMI score

Response: Yes BMI was calculated as part of the study procedure. In our paper, we recommend educating PLHIV about the negative effects of weight gain which they perceived to be favorable from the qualitative findings. Page 18 Line 400

8. Statement: The psychological impact of weight changes leading to non-adherence found in our study has been reported in other studies[7,12,20].

Comment: where is this reported and non-adherence to what?

Response: It was reported on page 12 (lines 245-253). Some participants reported that they would consider stopping treatment if they kept losing weight. We have clarified the treatment they would consider stopping which is antiretroviral treatment (line 247).

9. 9. Line 408 It is important to show the outcomes of similar studies in different settings or cultures and comment on these in the context of your findings.

Response:

The discussion section (pages 18-19) refers to studies conducted in other parts of South Africa like Cape Town (lines 415-422) and Durban (lines 411-414) as well as studies on weight perceptions conducted in other countries such as Tanzania (lines 406-409) and Uganda (lines 378-481).

10. How confident are the researchers that asking about a change of dress size can be used as proxy for weight loss or gain. what if they simply dont answer truthfully to this question? Are there studies that have shown dress size to be a reliable proxy for weight gain or loss? Studies

Response:

Since this was a qualitative study, we asked about perceptions of weight loss. Participants in our study reported that changes in dress sizes made them realize they lost weight. HCWs in our study recommended that asking participants changes in clothing size could be better than directly asking unintentional weight loss questions. Asking clothing size is a self-report measure which can get biased responses, but we feel is more acceptable to patients rather than directly asking about weight loss. A study in Netherlands found that self-reported clothing size appeared to predict cancer risk independently of body mass index, suggesting that self-reported clothing size can be a useful measure to consider in epidemiologic studies.

Hughes et al (2009) Self-reported clothing size as a proxy measure for body size

---

## [Decision Letter · Decision Letter 1]

12 Feb 2025

PONE-D-24-15618R1“If we lose it, we are worried”: Individual and provider level perceptions and attitudes towards weight change among people living with HIV screening for tuberculosis in routine health care settings in Gauteng Province, South Africa.PLOS ONE

Dear Dr. Dube,

Thank you for submitting your manuscript to PLOS ONE. After careful consideration, we feel that it has merit but does not fully meet PLOS ONE’s publication criteria as it currently stands. Therefore, we invite you to submit a revised version of the manuscript that addresses the points raised during the review process.

We look forward to receiving your revised manuscript.

Kind regards,

Metin Çınaroğlu

Academic Editor

PLOS ONE

Reviewers' comments:

Reviewer's Responses to Questions

**Comments to the Author**

1. If the authors have adequately addressed your comments raised in a previous round of review and you feel that this manuscript is now acceptable for publication, you may indicate that here to bypass the “Comments to the Author” section, enter your conflict of interest statement in the “Confidential to Editor” section, and submit your "Accept" recommendation.

Reviewer #1: All comments have been addressed

Reviewer #2: (No Response)

2. Is the manuscript technically sound, and do the data support the conclusions?

Reviewer #1: Yes

Reviewer #2: Partly

3. Has the statistical analysis been performed appropriately and rigorously? 

Reviewer #1: Yes

Reviewer #2: Yes

4. Have the authors made all data underlying the findings in their manuscript fully available?

Reviewer #1: Yes

Reviewer #2: Yes

5. Is the manuscript presented in an intelligible fashion and written in standard English?

Reviewer #1: Yes

Reviewer #2: No

6. Review Comments to the Author

Reviewer #1: The author has now addressed all the concerns earlier raised. The fact that the author has responded to each and every concern is quite encouraging. I believe no gray area has been left un answered.

Reviewer #2: Dear AUTHORS

thank you for submitting the draft. Whilst you have tried to address many of the comments i have a few concerns

1. there is an unacceptable high number of grammatical and language errors! This is a scientific journal and these issues must be addressed. For example, you cannot start a sentence with "Because" or say "In today's terms...."

2. Please read the title and think about my suggestion

3. The other concern I have is the discussion section where you compare or quote other studies ..please provide a bit more context to those studies so the reader has a sense of how similar or different the settings, context, methods, etc were.

There are a number of co-authors..i think you have sufficient capacity to address the grammar, language and other issues!

7. PLOS authors have the option to publish the peer review history of their article (what does this mean? ). If published, this will include your full peer review and any attached files.

**Do you want your identity to be public for this peer review?** For information about this choice, including consent withdrawal, please see our Privacy Policy .

Reviewer #1: **Yes: ** Dr. Hellen Jepngetich Keny

Reviewer #2: **Yes: ** Veerasamy Yengopal

---

## [Author Response · Author response to Decision Letter 2]

21 Jul 2025

Comment

We note that this data set consists of interview transcripts. Can you please confirm that all participants gave consent for interview transcript to be published?

If they DID provide consent for these transcripts to be published, please also confirm that the transcripts do not contain any potentially identifying information (or let us know if the participants consented to having their personal details published and made publicly available). We consider the following details to be identifying information:

- Names, nicknames, and initials

- Age more specific than round numbers

- GPS coordinates, physical addresses, IP addresses, email addresses

- Information in small sample sizes (e.g. 40 students from X class in X year at X university)

- Specific dates (e.g. visit dates, interview dates)

- ID numbers

Response:

Yes, the participants consented to having their de identified data shared in published journals.

The transcripts have been checked to remove any identifying information as recommended.

Comment 1

Line 1

“If we lose it, we are worried”: Individual and provider level perceptions and attitudes towards weight change among people living with HIV screening for tuberculosis in routine health care settings in Gauteng Province, South Africa.

please read this title again. The grammar seems incorrect...please check!

perhaps consider : perceptions and

attitudes towards weight change among people living with HIV who undergo TB screening

Response 1

Line 1, page 1

Agreed changed to

Individual and provider level perceptions towards weight change among people living with HIV who undergo TB screening in routine health care settings in Gauteng Province, South Africa.

Comment 2

Line 104 page 5

We were interested to know whether attitudes to weight and body size might affect how people responded to the questions in the WHO TB symptom screening tool and whether people with stavudine-related lipodystrophy might describe the change in body shape as weight loss.

Comment: This sentence should read “We sought to investigate whether…….

Line 103 page 5

Response 2

Agreed and changed to: We sought to investigate whether attitudes to weight and body size might affect how people responded to the questions in the WHO TB symptom screening tool and whether people with stavudine-related lipodystrophy might describe the change in body shape as weight loss.

Comment 3

Line 119, page 6

Tell us something more about the Focus groups...you say they were all male, all female and mixed. did you have a target group size? How many came from the XPHACTOR study and how many were invited from the ART support group? similarly, how many staff were art of the initial group and how many were invited? how did you determine sample size per group?

Response 3

Line 123-125 page 6

Adult PLHIV who were not XPHACTOR study participants were approached in-person for participation in the study via an ART support group linked to the clinic.

Lines 134 to 142

We initially planned to conduct around three FGDs with about eight to twelve participants in each group but eventually conducted seven FGDs to reach saturation. The number of discussions increased because we separated male and female FGD’s (two groups per gender) plus a mixed gender discussion to capture sufficient diversity in perspectives and experiences. Sample size per group was determined by availability of participants and previous qualitative research which recommends six to twelve participants per FGD(17,18). This range ensures sufficient diversity, a balanced interaction to allow all participants to contribute as well as effective facilitation of the discussion by the moderator.

Line 200-205 page 9

Seven FGDs (labelled 1-7) were held between 6th and 21st August 2014, with 7-12 participants in each focus group. The five adult PLHIV FGDs (two males, two females and one mixed gender) had 7-12 participants in each group. The HCWs and research staff FGD comprised of ten participants (seven routine HCWs and three research staff). The doctors and dieticians FGD had seven participants (five doctors and two dieticians) who were all routine staff.

Comment 4

Line 124, page 6

Adult PLHIV who were not XPHACTOR study participants were approached for participation in the study face to face via an ART support group linked to the clinic.

Comment: sentence should read as

"Adult PLHIV who were not XPHACTOR study participants were approached in-person for participation in the study via an ART support group linked to the clinic."

Line 123 page 7

Response 4

Sentence changed as suggested.

Adult PLHIV who were not XPHACTOR study participants were approached in-person for participation in the study via an ART support group linked to the clinic

Comment 5

Line 135, page 6

We initially planned to conduct around three FGDs but eventually conducted seven FGDs to reach saturation

Comment:

could you expand on this ....this related to the methodological approach taken:

Response 5

Line 136-142 page 7

The number of discussions increased because we separated male and female FGD’s (two groups per gender) plus a mixed gender discussion to capture sufficient diversity in perspectives and experiences. Sample size per group was determined by availability of participants and previous qualitative research which recommends six to twelve participants per FGD. This range ensures sufficient diversity, a balanced interaction to allow all participants to contribute as well as effective facilitation of the discussion by the moderator.

Comment 6

Line 161, page 8

All interviews were transcribed verbatim, and interviews done in local languages were translated to English

tell us more on this process and language translation is a complex process. one has to make sure that the nuances are not lost during the translation process!!

Response 6

Line 161-171 page 8

More details have been added:

Translations and transcriptions were done by experienced researchers fluent in the study languages. Transcriptions were done verbatim to keep nuances for instance, almost all deliberations were captured including non-verbal cues such as short silences and laughs. Quality assurance was conducted whereby a second researcher checked the accuracy of the transcripts against audio recordings.

Comment 7

Line 186, Page 9

could you add a few lines on the process to reach saturation?

Response 7

Line 193-196 page 9

TD, FM and JM reviewed and refined emerging themes repeatedly until saturation was achieved when no additional themes or categories could be identified.

Comment 8

Line 189, page 9

so there were main themes and sub-themes! the main themes identified are not what you are reporting in this sentence...they are sub-themes...separate them and be clear about the main and sub-themes!

Response 8

Line 206-211 page 10

The main themes and sub themes were separated as following;

We identified three main themes and five sub themes. The main themes were i. perceptions and attitudes towards weight change; ii ideal body size and shape; iii negative impact of weight loss on participant’s daily life. The sub themes were i) perceptions towards weight change; ii) attitudes towards weight loss; iii) stigma and loss of social support; psychological impact of weight loss; and iv) changes in dressing.

Comment 9

Line 375, page 17

please correct the grammar...you cannot start a sentence with "because"

Response 9

Line 393, page 18

Changed sentence to “As a result, the participants preferred gaining to losing weight”.

Comment 10

Line 378 page 17

Corroborating our finding, a study in Uganda found that women living with HIV did not want to lose weight because weight loss could lead to people suspecting their HIV seropositive status. Women living with HIV who lost weight in the same study also reported that they felt stigmatized, and it affected their adherence to ART[18]

This should read as "The results obtained in our study were similar to the findings of a Ugandan study that found that women living with HIV did not want to lose weight as this could lead to people suspecting that they were HIV positive.

Response 10

Line 397-401, page 18

Agreed and the sentence changed as suggested.

The results obtained in our study were similar to the findings of a Ugandan study that found that women living with HIV did not want to lose weight as this could lead to people suspecting that they were HIV positive

Comment 11

Line 381 page 17

The finding that it was rare for PLHIV to seek healthcare services because of weight loss concerns may negatively affect efforts to screen for and diagnose TB early.

did your study find this or was this from the literature?

Response 11

Line 401-404 page 18

This was a potential implication based on previous studies that have shown an association between loss of weight and delayed TB diagnosis/ seeking care.

The finding that it was rare for PLHIV to seek healthcare services because of weight loss concerns may negatively affect efforts to screen for and diagnose TB early. Several studies have found an association between weight loss and delay in TB diagnosis(21–23). A study consisting of 185 people with TB in KwaZulu Natal South Africa found that weight loss was associated with delayed care(21). The researchers suggested that stigma of weight loss because of its association with other diseases could have been a key factor contributing to this delay (21).

Comment 12

Line 388, page 17

A potential alternative to asking questions about weight loss when PLHIV present for treatment of check-ups is to measure their weight at the visit and compare this to previous weight measures recorded in their file.

Response 12

Line 413-415 page 19

Agreed and changed as suggested.

Comment 13

Line 389, page 17

The stigma associated with HIV and TB needs to be addressed for example by educating, through community outreach campaigns, both PLHIV, family members, and the community on HIV and TB using tool kits that have been developed to address TB-related stigma.

please correct the grammar ..this sentence does not make sense!

Line 426-428 page 9

Response 13

Agreed sentenced revised to:

The stigma associated with HIV and TB needs to be addressed for example by educating PLHIV, family members, and the community on HIV and TB. The education can be done through community outreach campaigns using tool kits that have been developed to address TB-related stigma

Comment 14

Line 397, page 18

A study conducted in Cape Town, South Africa found that participants preferred to be overweight and at risk of acquiring cardiovascular diseases, rather than to lose weight and be stigmatised as infected with HIV.

for better context, perhaps add a few lines about this study ...how many participants, what was the objective of this study, methodology used, etc.

Response

Line 424-440 page 19

Done, additional was added.

A mixed method study including 513 adult women was conducted in Cape Town, South Africa to explore the perception among black South African women that people who are thin are infected with HIV or have AIDS. The study found that participants preferred to be overweight and at risk of acquiring cardiovascular diseases, rather than to lose weight and be stigmatised as infected with HIV[5]

Comment 15

Line 410, page 18

Our study found that fuller women were considered to have the ideal body shape. This finding is consistent with findings from other previous studies.

Comment

Add references:

Response 15

References cited. Line 440

Matoti-Mvalo T, Puoane T. Perceptions of body size and its association with HIV/AIDS. South African Journal of Clinical Nutrition. 2011;24(1):40–5.

Nyamaruze P, Govender K. I like the way i am, but i feel like i could get a little bit bigger”: Perceptions of body image among adolescents and youth living with HIV in Durban, South Africa. PLoS One [Internet]. 2020;15(1):1–10. Available from: http://dx.doi.org/10.1371/journal.pone.0227583

Comment 16

Line 431, page 19

Our study findings are significant in today’s terms as TB remains a leading cause of death among PLHIV

Delete “in today’s terms” in the sentence

Response 16

Deleted as suggested. Line 462

Our study findings are significant as TB remains a leading cause of death among PLHIV so understanding their perceptions around weight change and how this may affect TB symptom screening is important

Comment 17

The other concern I have is the discussion section where you compare or quote other studies ..please provide a bit more context to those studies so the reader has a sense of how similar or different the settings, context, methods, etc were.

Response 17

Additional information has been added to the studies cited in the discussion.

Lines 397, 404, 424, 436, 441, 458

Comment 18

there is an unacceptable high number of grammatical and language errors! This is a scientific journal and these issues must be addressed. For example, you cannot start a sentence with "Because" or say "In today's terms...."

Response 18

Incorrect grammar has been revised in the manuscript

Lines 89, 104, 144, 168, 216, 261, 264, 298, 358, 393, 406, 454, 457, 458, 484

---

## [Decision Letter · Decision Letter 2]

24 Aug 2025

“If we lose it, we are worried”: Individual and provider level perceptions towards weight change among people living with HIV who undergo TB screening in routine health care settings in Gauteng Province, South Africa.

PONE-D-24-15618R2

Dear Dr. Dube,

We’re pleased to inform you that your manuscript has been judged scientifically suitable for publication and will be formally accepted for publication once it meets all outstanding technical requirements.

Kind regards,

Dorina Onoya

Academic Editor

PLOS ONE

Additional Editor Comments (optional):

Reviewers' comments:

Reviewer's Responses to Questions

**Comments to the Author**

1. If the authors have adequately addressed your comments raised in a previous round of review and you feel that this manuscript is now acceptable for publication, you may indicate that here to bypass the “Comments to the Author” section, enter your conflict of interest statement in the “Confidential to Editor” section, and submit your "Accept" recommendation.

Reviewer #1: (No Response)

Reviewer #2: All comments have been addressed

2. Is the manuscript technically sound, and do the data support the conclusions?

Reviewer #1: Partly

Reviewer #2: Yes

3. Has the statistical analysis been performed appropriately and rigorously? 

Reviewer #1: Yes

Reviewer #2: N/A

4. Have the authors made all data underlying the findings in their manuscript fully available?

Reviewer #1: No

Reviewer #2: Yes

5. Is the manuscript presented in an intelligible fashion and written in standard English?

Reviewer #1: Yes

Reviewer #2: Yes

6. Review Comments to the Author

Reviewer #1: Your statement on data availability is still not clear. Please adhere to the Journal's data policy provisions

Reviewer #2: I am pleasantly surprised by this draft. the authors have realised the importance of good language and grammar. It seems like they sent this for English editing which is good. i can see the significant improvement in the quality of the language, grammar and content!

7. PLOS authors have the option to publish the peer review history of their article (what does this mean? ). If published, this will include your full peer review and any attached files.

**Do you want your identity to be public for this peer review?** For information about this choice, including consent withdrawal, please see our Privacy Policy .

Reviewer #1: **Yes: ** Hellen Jepngetich

Reviewer #2: **Yes: ** veerasamy yengopal

---

## [Editor Report · Acceptance letter]

PONE-D-24-15618R2

PLOS ONE

Dear Dr. Dube,

I'm pleased to inform you that your manuscript has been deemed suitable for publication in PLOS ONE. Congratulations! Your manuscript is now being handed over to our production team.

Kind regards,

on behalf of

Dr. Dorina Onoya

Academic Editor

PLOS ONE